# Development and psychometric properties of the Digital Difficulties Scale (DDS): An instrument to measure who is disadvantaged to fulfill basic needs by experiencing difficulties in using a smartphone or computer

**Sarah Anrijs** [ID]*[☯], **Koen Ponnet** [ID][☯], **Lieven De Marez**

Faculty of Social Sciences, imec-mict-ugent, Ghent University, Ghent, Belgium

☯ These authors contributed equally to this work.
* sarah.anrijs@ugent.be

**Data Availability Statement:** All data files are available from the openICPSR database (accession: https://doi.org/10.3886/E118021V1).

## Abstract

Today, some individuals may be at a disadvantage by experiencing difficulties in using a smartphone or computer to reach specific outcomes (e.g., looking for a job, searching for information on insurances) or in general (e.g., not knowing how to change the settings of an app or website). The aim of this study is to develop and examine the psychometric properties of a new instrument, called the Digital Difficulties Scale (DDS). A multi-phase method was performed to develop the questionnaire in the period from January 2019 to November 2019. The item pool was generated based on a literature review, informal observations and interviews. Then, this item pool was presented both to experts ($n = 6$) and non-experts ($n = 492$) to assess content and face validity. In a second stage, construct validity (both exploratory and confirmatory), convergent and divergent validity, internal consistency, and test-retest reliability of the questionnaire were tested. These analyses were based on a representative sample ($n = 1000$), and an independent sample for test-retest reliability ($n = 44$). Twenty-four items were generated and refined during content and face validity assessment. The exploratory factor analysis revealed three factors (Specific Digital Difficulties, General Digital Difficulties, and Worries about Future Digital Difficulties) containing sixteen items, together explaining 73.03% of the observed variance. The confirmatory factor analysis proved adequate model fitness. Both convergent and divergent validity were good, and internal consistency was excellent, with Cronbach's alphas ranging between .93 and .97. Finally, our instrument demonstrated good test-retest reliability, with interclass correlation coefficients between .73 and .86. Consequently, the DDS can be used both in future research and practice, as it is a valid and reliable instrument to measure who is disadvantaged to fulfill basic needs by experiencing difficulties in using a smartphone or computer.

**Funding:** Author SA received a research grant from Ghent University (Bijzonder Onderzoeksfonds) with grant number 01D33218. The funder had no role in study design, data collection and analysis, decision to publish, or preparation of the manuscript.

**Competing interests:** The authors have declared that no competing interests exist.

## Introduction

It is often taken for granted that every individual is able to use a smartphone or computer [1]. This normalization of the use of smartphones and computers is reflected in many workplaces, and in the services of commercial and public organizations [2]. For instance, while in the past offline services could be consulted to make financial transfers, to claim a benefit, or to search for job vacancies, more and more organizations are shifting their offline services towards online alternatives with less or no face-to-face support for the individual [3,4]. In the workplace, programs and online tools for communication, time registration, project management, and data storage have become common practice nowadays, including regular updates and the introduction of new digital tools [5]. Although, the use of digital services and programs may increase efficiency or simplify processes [2,6,7], the question raises whether some individuals may be at a disadvantage to fulfill basic needs due to these changes, as persons may experience difficulties in using a smartphone or computer to look for a job, to accomplish job tasks, to search for information on insurances, or to contact the government [8].

The unequal access to or adoption of smartphones, computers, and internet is the focus of digital inequality research or digital divide research [8,9]. Research within this domain has repeatedly demonstrated that a significant amount of individuals lack access, knowledge, positive attitudes, skills or support to adopt a smartphone, computer or internet in general as well as for specific uses, such as using a search engine, using a word processor, making new contacts online, or buying a product online, both in developed and developing countries, in disabled and non-disabled populations, among poor and non-poor, and across old and young individuals [6,7,10–21].

Although several scales exist in the domain of digital inequality research, it is our view that at the moment, no comprehensive validated scale exists that measures who is disadvantaged by encountering digital difficulties or experiencing questions and problems in using a smartphone or computer. Existing instruments in the domain of digital inequality rather measure individuals' levels of digital proficiency or literacy than their level of digital difficulties, as they focus on an individual's perceived capability to perform a broad range of skills, on variety in use, or on acquired outcomes [1,22]. For example, existing scales measure if individuals know how to refresh a button, how to set a bookmark, or how to put a video online, or they measure how frequently individuals surf to a friend's blog, read information about raising children, or buy products online [e.g., 15,23–25]. The measurement of such skills and uses may be interesting to investigate digital inequality in specific contexts, however, this goes beyond the question who may be at a disadvantage to fulfill basic needs as a result of experiencing difficulties in using a smartphone or computer. More specifically, individuals are at a disadvantage, if they are unable to use a smartphone or computer for basic needs which are (likely to be) exclusively reachable through online services. In line with social exclusion measures, basic needs refer to needs that are related to income, housing, and healthcare [26,27]. For example, if a government decides that unemployment benefits can only be claimed through an online form, having difficulties in using a smartphone or computer can result into not receiving this benefit, which may lead to having too little money to pay the house rent.

Therefore, the purpose of the present study is to develop a straightforward and easy-to-administer instrument, called the Digital Difficulties Scale. Additionally, the second aim of this study is to evaluate the psychometric properties of the Digital Difficulties Scale in terms of validity and reliability. In order to establish the validity of our instrument, logic, construct, convergent and divergent validity will be examined. Reliability can be assured by demonstrating good internal consistency and test-retest results. If the Digital Difficulties Scale establishes sound psychometric properties, the instrument can be used by researchers, practitioners and

policy makers (1) to understand who encounters digital difficulties and therefore may be disadvantaged in digitizing societies and (2) to set up interventions that help individuals to overcome the difficulties they experience in using a smartphone or computer.

## Development and assessment of the Digital Difficulties Scale

One of the greatest challenges in conducting survey research is assuring the accuracy of measurement of the examined constructs [28,29]. As valid and meaningful conclusions can only be drawn from valid and reliably measurement, adequate measurement is crucial and necessary in every survey research, irrespectively of the applied analytic techniques [29,30]. Developing valid and reliable scales is a time-consuming and hard process, in which the ability to accurately operationalize the unobservable construct and the ability to critically evaluate the added value of each item are highly important. The process of item generation and validity assessment should be clearly reported, in order to assure transparency [29].

Following these recommendations, the present study was conducted in two stages. In a first stage, we started with item generating, following a combined deductive and inductive approach. Afterwards, logic validity was assessed and the most appropriate phrasing was determined. In the second stage, the psychometric properties of the scale were evaluated. For this purpose, exploratory and confirmatory factor analysis was conducted in order to examine the construct validity. Furthermore, our instrument was evaluated in terms of convergent and divergent validity, and reliability. Below, we elaborate in detail on both stages. The study protocol was approved by the Ethics Committee of the Faculty of Political and Social Sciences of Ghent University and informed consent was obtained of all participants before completing the questionnaire. All data were analyzed anonymously.

### Stage 1: Item generation and scale development

**Item generation.**    The starting point of our item generation was the Internet Outcomes Scale (IOS) of Van Deursen and Helsper [24], two example items of this scale are "through the internet I found a (better) job" and "through the internet I met a potential partner using online dating". The IOS has proved to be useful to answer several research questions as it has been adopted by several researchers [e.g., 11,31]. Although the IOS is not able to measure which individuals are disadvantaged due to difficulties they encounter in using a smartphone or computer, some items of the IOS were a good starting point because they encompass specific outcomes (e.g., finding a job, finding medical information, contacting the government). Departing from social exclusion research that defines basic needs as needs that are related to income, housing and healthcare [26,27], the IOS helped us to create a first set of eleven items for our scale, an example item was, "I have had difficulties to arrange paper work or payments because of my limited computer, smartphone or internet access or skills (such as health insurances, taxes, electricity bills)".

In addition to the items that were deductively derived from the IOS and the definition of basic needs, additional items were inductively constructed based on informal observations and short interviews, conducted by the first and second author. For instance, we observed and informally interviewed family members while using their smartphone or computer, and volunteered to give support and advice on computer problems in a public library for six months. More specifically, based on these observations and interviews, the question raised whether some individuals might also be at a disadvantage not because of having difficulty in reaching specific outcomes online, but rather because they struggle with using a smartphone or computer in general and/or because they feel a sense of insecurity thinking about the use of digital technologies in the (near) future. This consideration came to us because we noticed during the

observations and interviews that some individuals very frequently experience general problems or questions in using a smartphone and computer, are highly dependent on others, or feel quite insecure using a smartphone or computer now and in the future. Therefore, we included thirteen additional items in our instrument. Five out of these thirteen items were adapted from a study of Cassidy and Eachus [32], two example items are: "In general, I need help when trying something new on a smartphone, computer, or the internet", "In general, I am able to solve questions or problems that I encounter when using a smartphone, computer or the internet myself". Two items were self-created based on the above-mentioned observations. These items were: "In general, I find updates in a smartphone, computer or the internet frustrating (for example, the navigation menu has changed)", and "In general, I find it difficult to change settings in a smartphone, computer, or the internet (for example, privacy or safety settings)". Six items were inspired by the financial insecurity construct as operationalized in the Financial Stress Scale of Ponnet et al. [33], two example items are: "I am often worried that in the future I will not be able to keep up with changes in smartphones, computers or the internet", and "I am frightened that in the future smartphones, computers or the internet will be too complicated to use for me". These six items reflect a sense of discomfort thinking about the use of digital technologies in the (near) future.

In conclusion, 24 items were initially included in the Digital Difficulties Scale measuring who is disadvantaged to fulfill basic needs by experiencing difficulties in using a smartphone or computer. These 24 items reflected difficulties to reach specific outcomes online, difficulties that individuals experience in using a smartphone and computer in general, and senses of discomfort or insecurity about the use of smartphones and computers in the future.

**Scale development.** In this phase, the 24-item scale was evaluated on logic validity. The assessment of logic validity is an important step in scale development. It is applied in order to assess the relevance and added value of each item in relation to the unobservable concept, and to evaluate the clarity and parsimony of item wording [29]. Logic validity encompasses both content validity and face validity, the former is evaluated by experts, the latter is examined by non-experts [29].

*Content validity*. Content validity was assessed in a qualitative way, consulting a scientific expert panel of six members (i.e., a team of academic researchers from [name deleted for purpose of blind review] specialized in social exclusion, social indicators, and policy assessment). The expert panel assessed the relevance, the wording, grammar, item allocation, and scaling of each item. Based on the feedback of the experts several changes were made. For instance, the wording of one reversed-scored item was reformulated in the same direction as the other items (e.g., "In general, I am able to solve questions or problems that I encounter when using a smartphone, computer or the internet myself" was changed to "In general, I am not able to solve questions or problems that I encounter when using a smartphone, computer or the internet myself"), as several panel members agreed that the first formulation could be confusing for respondents. Indeed, it is recommended to avoid reversed-scored items, as these often reduce the validity of questionnaire responses and can introduce systematic errors [29]. Furthermore, three items related to difficulties to reach specific outcomes online were considered as irrelevant or formulated too abstractly, and therefore were omitted (e.g., "I have had difficulties to fulfill certain job expectations because of my limited computer, smartphone or internet access or skills"). The experts also suggested to split some other items related to difficulties in specific outcomes, in order to make them more concrete and unambiguously both for respondents, researchers and practitioners (e.g., "I have had difficulties to arrange paper work or payments because of my limited computer, smartphone or internet access or skills (such as, health insurances, taxes, electricity bills)" was reformulated to "To what extent do you have difficulty in using a smartphone, computer or the internet to arrange payments", "To what extent do you

**Table 1. Characteristics of the study samples.**

| | Sample 1 (n = 299) Number (%) | Sample 2 (n = 193) Number (%) | Sample 3 (n = 1000) Number (%) | Sample 4 (n = 44) Number (%) |
|---|---|---|---|---|
| Gender | | | | |
| Men | 139 (46.50) | 89 (46.10) | 502 (50.20) | 15 (34.10) |
| Women | 160 (53.50) | 104 (53.90) | 498 (49.80) | 29 (65.90) |
| Age | | | | |
| 18–34 years | 90 (30.10) | 61 (31.60) | 332 (33.20) | 4 (9.10) |
| 35–49 years | 66 (22.10) | 54 (28.00) | 345 (34.50) | 20 (45.50) |
| 50–64 years | 75 (25.10) | 40 (20.70) | 323 (32.30) | 16 (36.40) |
| 65 years and above | 68 (22.70) | 38 (19.70) | / | 4 (9.10) |
| Mean (SD) | 47.61 (17.44) | 45.66 (17.22) | 41.47 (13.35) | 49.16 (10.83) |
| Range | 18–84 | 18–88 | 18–64 | 24–70 |
| Employment status | | | | |
| Employed | 142 (47.50) | 107 (55.40) | 751 (75.10) | 32 (72.70) |
| Unemployed | 157 (52.50) | 86 (44.60) | 249 (24.90) | 12 (27.30) |
| Educational level | | | | |
| No, primary or lower secondary education | 17 (5.70) | 7 (3.60) | 204 (20.40) | 0 (0.00) |
| Secondary education | 79 (26.40) | 48 (24.90) | 390 (39.00) | 4 (9.10) |
| Higher education | 203 (67.90) | 138 (71.50) | 406 (40.60) | 40 (90.90) |
| Mother was born in Belgium | | | | |
| Yes | 254 (84.90) | 179 (92.70) | 895 (89.50) | 44 (100.00) |
| No | 45 (15.10) | 14 (7.30) | 105 (10.50) | 0 (0.00) |

have difficulty in using a smartphone, computer or the internet to complete your tax return", "To what extent do you encounter difficulties in using a smartphone, computer or the internet to find information on health insurances"). After consulting the expert panel, our pre-final instrument consisted of 23 items. Answers ranged from 1 = *totally disagree/having no difficulty* to 6 = *totally agree/having difficulty*. The items related to having difficulty to reach specific outcomes using a smartphone, computer or the internet could also be answered with 7 = *does not apply for me*.

*Face validity*. Face validity was examined based on two cross-sectional studies with non-experts. In the first study, a convenience sample of 299 individuals completed our 23-items instrument. The respondents were recruited during five days in a public library with more than 7000 visitors a day. Table 1 presents the descriptive characteristics of the 299 respondents (see sample 1). Consistent with Khazaee-Pool et al. [34], after completion respondents were asked if they felt ambiguity or problems in replying the questionnaire, and whether or not something was missing in the instrument. In addition, we conducted individual semi-structured interviews with elderly persons (n = 5), persons with migration background (n = 3), and persons with cognitive disability (n = 2). During these interviews, respondents were asked to evaluate each item of the scale separately in presence of the first author in order to gather more in-depth information about the clarity and simplicity of item wording and scaling. Based on notes that were taken of the comments from survey respondents and interviewees, the scale was fine-tuned in the following ways:

First, with regard to the thirteen items reflecting inconvenience when using a smartphone or computer in general, and/or feeling a sense of insecurity thinking about using digital technologies in the (near) future, the wording of the insecurity-items, were changed with stronger emphasize on the aspect of worries, as the items were considered as ambiguous. In order to do

so, we used the wording of an existing worry-scale of Tallis, Eysenck, and Mathews [35] (e.g., "How often do you worry that in the future you will be unable to keep up with ongoing changes in smartphones, computers or the internet" instead of "I am afraid that in the future I will be unable to keep up with ongoing changes in smartphones, computers or the internet"). Furthermore, the original six insecurity-items were reformulated to three worry-items as respondents did not notice substantial difference between all six items. For the same reason, two out of the five items that were adapted from Cassidy and Eachus [32] were removed (i.e., "In general, I find it difficult to let a smartphone, computer or the internet do what I want it to do" and "In general, if something goes wrong when using my smartphone, apps, websites or computer, I do not know why that is"). As pointed out by Cronbach and Meehl [36], item parsimony is important to obtain a valid and reliable instrument. Second, the wording of several items related to difficulties to reach specific outcomes was made more concrete with additional examples. Also, the preamble and the answer options of these items were adapted. More specifically, the preamble was changed into "If necessary, to what extent would you have difficulty to reach the following outcomes, without help of others. Answer options were 1 = *having no difficulty*, 2 = *having rather no difficulty*, 3 = *having rather difficulty*, and 4 = *having difficulty*. Third, all phrases that contained "using a smartphone, computer or the internet" were replaced by "using my smartphone, apps, websites or computer programs" or "online", as the former was considered as too abstractly formulated. After these changes, our instrument consisted out of 18 items.

In order to evaluate the refined version of our scale, a second cross-sectional study was conducted with non-experts. A new convenience sample of 193 individuals was recruited in a library, one month after the first sample data collection. Descriptive characteristics of this sample are provided in Table 1 (see sample 2). Again, respondents were asked whether all items were clear, simple, and relevant. Respondents' feedback was largely positive. Based on the feedback we decided to add one extra item, i.e., "If necessary, to what extent would you have difficulty to apply for jobs online, without help of others (e.g., uploading your cv or motivation letter)". Thus, the final test version of our instrument consisted out of 19 items assessing individuals' perceived difficulties in using a smartphone or computer. We refer to S1 Appendix for an overview of the items.

## Stage 2: Validity and reliability assessment

The aim of this stage was to assess construct validity using exploratory factor analysis (EFA) and confirmatory factor analysis (CFA), convergent and divergent validity, and reliability of the Digital Difficulties Scale in a wider setting. Therefore, the scale was administered in a new sample of Flemish individuals (i.e., the Dutch speaking part of Belgium) between 18 and 64 years old. Respondents were recruited by a professional research organization that has access to a panel of 300.000 Belgian individuals. Students and non-Dutch speaking persons were not eligible to participate. In total, 8000 panel members received an email with a short study description, informed consent, and invitation to participate, including a link to the survey. A stratified sampling procedure was applied in order to assure that the sample was heterogeneous. Based on the federal statistics of Belgium (www.statbel.fgov.be), we a priori stratified the data with regard to gender, age, employment status and educational degree, so that the proportion of the strata reflects the proportion of the Flemish population. At the moment that 1000 respondents had been reached in accordance with the interlaced strata, the survey link was closed by the professional research organization. Descriptive characteristics of the sample are provided in Table 1 (see sample 3). First, construct validity was tested using both exploratory and confirmatory factor analysis. Second, convergent and divergent validity, internal

consistency and test-retest reliability were assessed in order to evaluate the newly developed scale. Below, we first describe the analysis methods, followed by the results of this stage.

**Analysis.** *Construct validity*. EFA is applied to specify the main factors of our instrument. The minimum recommended sample size is five individuals per item or 95 respondents, following Gable and Wolf [37]. To assess the adequacy of the sample for the factor analysis, the Kaiser-Meyer-Olkin (KMO) measure and Bartlett's test of sphericity should be consulted. KMO values of .60 or higher indicate an acceptable sample, values between .80 and 1 indicate an adequate sample [38]. Bartlett's test of sphericity tests the hypothesis of an unrelated correlation matrix, which is unsuitable for factor analyses as no structure could be detected in such a case. P-values of less than .05 indicate that factor analysis is useful to apply on the data [39]. Any factor with an eigenvalue above 1 is considered significant for factor extraction, factor loadings equal to or greater than .40 are considered as acceptable [40]. However, the present study aimed for factor loadings equal to or greater than .50, pursuing adequate to strong factor loadings [41]. Afterwards, CFA is applied in order to assess the coherence between the data and the factor-structure derived from EFA. Following Soper [42], the recommended minimum sample size for our model structure is 256, the minimum sample size to detect effect is 119, this calculation is based on an anticipated effect size of .30, desired statistical power level of .80, and probability level of .05, 3 latent variables, and 19 observed variables. The model fit should be assessed using several fit indices, including Chi-square, Comparative Fit Index (CFI), Tucker Lewis Index (TLI), Root Mean Square Error of Approximation (RMSEA), and Standardized Root Mean Square Residual (SRMR). Both CFI and TLI range from 0 to 1, values of .95 or higher demonstrate a good model fit, values of .90 indicate an adequate model fit [43,44]. An RMSEA value between .08 and .10 demonstrates an average fit, values between .06 and .08 indicate an adequate fit, and values below .05 show a good model fit [45,46]. SRMR values smaller than .08 and .05 indicate a relatively good and a good model fit, respectively [44]. Consistent with other studies [47,48], we decided a priori that—if indicated by modification indices—correlated error terms were allowed across similarly worded items in order to rule out response bias.

*Convergent and divergent validity*. Item-convergent validity is examined by calculating the correlations between each item score and identified subscales scores of the Digital Difficulties Scale. Using Pearson correlations, item-convergent validity is met when for each identified subscale, each item of the subscale significantly correlates more with the total score of its respective subscale, rather than with the total score of other subscales [34]. Correlation coefficients between 0 and .20 are considered poor; fair between .21 and .40; good between .41 and .60; very good between .61 and 80; and excellent above .81 [49]. Convergent and divergent validity is assessed on subscale-level, using existing validated measures. Convergent validity is accepted when a subscale correlates positively with the validated measure (i.e., correlation coefficient .21 or above). Divergent validity is established when a negative moderate correlation coefficient appears between a subscale and a validated measure (i.e., correlation coefficient of -.21 or lower) [34]. Below, the selected validated scales to assess convergent and divergent validity are described.

The Technology Readiness Index (TRI). The TRI measures individuals' propensity to embrace and use new technologies [50] and consists of four subscales (i.e., Optimism, Innovativeness, Discomfort, and Insecurity). The TRI has been used in several studies in which it shows to have good internal reliability and a stable factorial structure [e.g., 51,52]. The present study included all four items from the Innovativeness subscale (i.e. the tendency to be a technology pioneer) and three items from the Discomfort subscale (i.e. the perceived lack of control over technology and a feeling of being overwhelmed by it). One item of the original Discomfort subscale was not adopted as according to the authors of the original study this

item rather focuses on distrust than on discomfort. Each item is scored on a five-point Likert scale, ranging from 1 = *totally disagree* to 5 = *totally agree*. The minimum and maximum score is 4 and 20 for Innovativeness, and 3 and 15 for Discomfort. A higher score on Innovativeness indicates higher readiness for new technologies, and a higher score on Discomfort suggests lower readiness for new technologies. In our sample, the Cronbach's alpha coefficient was .84 and .85 for Innovativeness, and Discomfort, respectively, indicating good reliability.

The Internet Skills Scale (ISS). The ISS is a Dutch validated questionnaire consisting of four subscales (i.e., Operational Skills, Information Navigation, Social Skills, and Creative Skills) with each five items [25]. The questionnaire captures individuals' perceived ability to use a browser, to search information online, to communicate online, and to create online content, from basic to advanced levels. In this study, we included all five items of the Information Navigation subscale. Items were scored on a five-point Likert scale, ranging from 1 = *totally disagree* to 5 = *totally agree*. Higher scores indicate lower online searching and navigating skills. The minimum and maximum scores of this scale are 5 and 25. The internal consistency of the Information Navigation subscale was good, with Cronbach's alpha .85.

*Reliability. internal consistency and test-retest reliability*. Scale reliability of the DDS can be examined on two criteria, i.e., internal consistency and test-retest reliability. The internal consistency of the subscales and the total scale was conducted with sample 3 (see Table 1 for the descriptives). Cronbach's alpha values equal to or higher than .70 are considered modest but acceptable, values higher than .80 are good, and values higher than .90 are excellent [49,53]. The test-retest reliability was conducted with an independent sample of 44 individuals, with a two to three weeks interval between the first and second administration. The descriptives of the sample can be found in Table 1 (sample 4). For each subscale and the total scale the intra-class correlation coefficient (ICC) was estimated. ICC values of .40 or above are considered acceptable [54].

All statistical analyses were performed using SPSS 25.0, except CFA, which was conducted using Mplus 8.

## Results

### Construct validity

First, we conducted an EFA using principal axis factoring with varimax rotation. The KMO-measure was .927 and the Barlett's test of sphericity was significant ($\chi^2$ (171) = 18533.24, p < .001), demonstrating that our main sample is adequate for EFA. Initially, four factors showed eigenvalues above 1 for the nineteen-item instrument, accounting for 71.56% of observed variance. However, based on assessment of the item loadings, three items showed factor loadings below .50 (i.e., SDD9, SDD10, and SDD11) and therefore were step-by-step removed using iterative factor analysis. This process resulted in a good three-factor solution for sixteen items, explaining 73.03% of variance. As expected, these three factors were: Factor 1 (Specific Digital Difficulties (SDD)) including eight items and accounting for 32.44% of explained variance, Factor 2 (General Digital Difficulties (GDD)) including five items explaining 23.83% of observed variance, and Factor 3 (Worries about Future Digital Difficulties (WFDD)) including three items explaining 17.76% of observed variance. Table 2 provides the factor loadings of the items.

Next, a CFA was conducted on the sixteen items in order to test the instrument's model fitness. As our data were non-normally distributed, results were obtained with the maximum likelihood mean adjusted. Modification indices suggested to allow three error covariance between closely related items. The results of the fit statistics indicate an adequate model fit: $\chi^2$(98) = 498.61, *p* < .001, CFI = .97, TLI = .96, RMSEA = .064 (CI .058 - .070), SRMR = .048. All factor loadings were above .62 or above. Fig 1 shows the model.

**Table 2. Exploratory factor analysis of the Digital Difficulties Scale (sample 3, n = 1000).**

| Item | Factor 1—SDD | Factor 2—GDD | Factor 3—WFDD |
|---|---|---|---|
| SDD 4 | **.818** | .217 | .132 |
| SDD 6 | **.807** | .161 | .175 |
| SDD 7 | **.799** | .158 | .142 |
| SDD 5 | **.795** | .205 | .151 |
| SDD 3 | **.790** | .226 | .177 |
| SDD 2 | **.722** | .236 | .149 |
| SDD 8 | **.713** | .201 | .182 |
| SDD 1 | **.671** | .228 | .158 |
| GDD 2 | .243 | **.808** | .263 |
| GDD 3 | .218 | **.808** | .228 |
| GDD 5 | .250 | **.792** | .222 |
| GDD 4 | .234 | **.766** | .261 |
| GDD 1 | .273 | **.732** | .275 |
| WFDD 2 | .247 | .366 | **.866** |
| WFDD 3 | .259 | .377 | **.856** |
| WFDD 1 | .265 | .378 | **.826** |

SDD = Specific Digital Difficulties, GDD = General Digital Difficulties, WFDD = Worries about Future Digital Difficulties.

Figures in bold are related to factor loadings equal to or greater than .671.

## Convergent and divergent validity

Item-convergent validity for the Digital Difficulties Scale is presented in Table 3. All coefficients are higher than .77, and most of them are higher than .81, indicating very good to excellent correlation coefficients. The subscale Worries about Future Digital Difficulties had the highest item-convergent validity, the subscale Specific Digital Difficulties had the lowest item-convergent validity. Scale-convergent and divergent validity was assessed by correlations between the different subscales of the Digital Difficulties Scale and items from the TRI and the ISS. More specifically, we expected a moderate positive correlation between General Digital Difficulties and the TRI—Discomfort subscale, and a negative correlation between General Digital Difficulties and the TRI—Innovativeness subscale. Specific Digital Difficulties was only tested on convergent validity, based on correlations with the ISS–Information Navigation subscale. No existing validated scales were suitable to evaluate convergent and divergent validity of the Worries about Future Digital Difficulties subscale. The correlation coefficients between Specific Digital Difficulties and ISS—Information Navigation, and between General Digital Difficulties and TRI—Discomfort were respectively .43 and .52, indicating a good convergent validity for both subscales. The correlation between General Digital Difficulties and TRI—Innovativeness was -.52, indicating that the divergent validity for this subscale was good (see Table 4).

## Reliability: Internal consistency and test-retest reliability

In order to assess the internal consistency of the Digital Difficulties Scale, Cronbach's alpha was calculated. Cronbach's alpha coefficients for the subscales Specific Digital Difficulties, General Digital Difficulties, and Worries about Future Digital Difficulties were respectively .94, .93, and .97. Cronbach's alpha for the total instrument was .94, indicating excellent

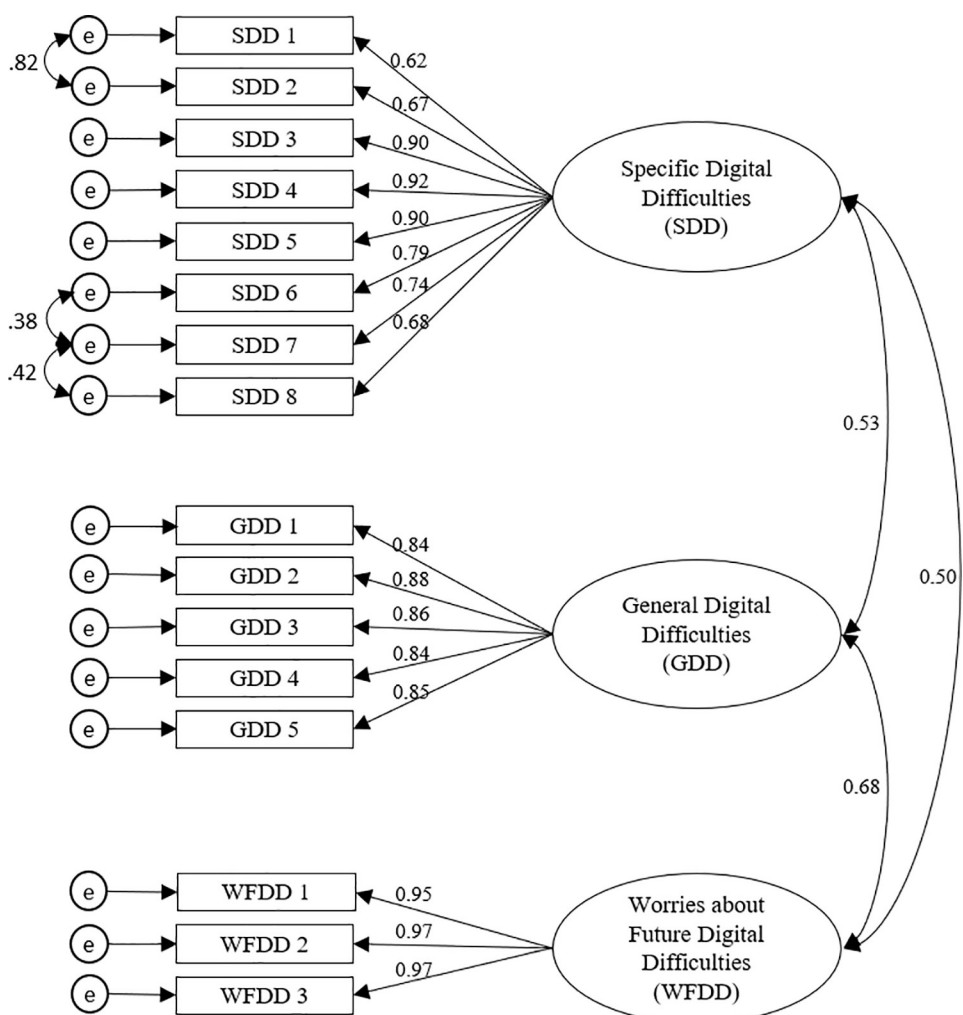

**Fig 1. Measurement model.** The three-factor model obtained from confirmatory factor analysis (sample 3, n = 1000).

internal reliability. The stability of the total instrument was evaluated by conducting a test-retest reliability analysis. ICC was .84 for the total scale, .73 for Specific Digital Difficulties, .86 for General Digital Difficulties, and .77 for Worries about Future Digital Difficulties, indicating satisfactory to good test-retest reliability.

## Discussion

Considering that experiencing difficulties in using a smartphone or computer may place individuals at a disadvantage to fulfill basic needs, the present study describes the development and psychometric properties of the Digital Difficulties Scale (DDS) to measure this form of inequality. Initial items were generated during eight months, based on a thorough review of the literature, and informal observations and interviews. In the scale development phase, expert consultation was used to ensure that this measure had both theoretical and practical value with regard to digital inequality research and policy decisions. Items were further refined and selected based on input from non-experts in two smaller cross-sectional studies. In the testing phase, our scale was evaluated on construct validity, convergent and divergent validity,

**Table 3. Item-convergent validity: Item-scale correlation matrix for the Digital Difficulties Scale (sample 3, n = 1000).**

| | | SDD | GDD | WFDD |
|---|---|---|---|---|
| SDD (item number) | | | | |
| | SDD 1 | **.837** | .402 | .382 |
| | SDD 2 | **.847** | .403 | .370 |
| | SDD 3 | **.787** | .440 | .426 |
| | SDD 4 | **.817** | .429 | .388 |
| | SDD 5 | **.800** | .418 | .393 |
| | SDD 6 | **.803** | .395 | .391 |
| | SDD 7 | **.799** | .369 | .358 |
| | SDD 8 | **.767** | .397 | .375 |
| GDD (item number) | | | | |
| | GDD 1 | .430 | **.842** | .563 |
| | GDD 2 | .448 | **.900** | .583 |
| | GDD 3 | .411 | **.900** | .551 |
| | GDD 4 | .431 | **.879** | .573 |
| | GDD 5 | .454 | **.891** | .552 |
| WFDD (item number) | | | | |
| | WFDD 1 | .450 | .620 | **.965** |
| | WFDD 2 | .444 | .621 | **.977** |
| | WFDD 3 | .446 | .629 | **.974** |

SDD = Specific Digital Difficulties, GDD = General Digital Difficulties, WFDD = Worries about Future Digital Difficulties.

Bold data reflect higher item-scale correlation for the three factors of the Digital Difficulties Scale.

and internal consistency, based on a representative sample. Finally, test-retest reliability was assessed using an independent sample of 44 individuals.

The findings of this study indicate that the psychometric properties of our instrument are good. The results of EFA and CFA demonstrated a good factorial structure for a 16-item instrument. The EFA revealed that 73.03% of the total observed variance could be explained by a three-factor structure of the instrument: Specific Digital Difficulties, General Digital Difficulties, and Worries about Future Digital Difficulties. Furthermore, the CFA demonstrated acceptable fit indices for the three-factor model. The convergent and divergent validity was good both on item and subscale level. Finally, the Cronbach's alpha coefficients indicate an excellent reliability of our scale. Interclass correlation coefficients indicated good test-retest reliability for the Digital Difficulties Scale and the subscales.

**Table 4. Convergent and divergent validity: Correlations between two subscales of the Digital Difficulties Scale and other validated questionnaires (sample 3, n = 1000).**

| | Internet Skills Scale - Information Navigation | Technology Readiness Index—Discomfort | Technology Readiness Index—Innovativeness |
|---|---|---|---|
| SDD | **.433** | .336 | -.331 |
| GDD | .561 | **.514** | **-.515** |

SDD Specific Digital Difficulties, GDD General Digital Difficulties.

The bold data reflect moderate to good correlations between the subscales of the Digital Difficulties Scale and other validated questionnaires.

The Specific Digital Difficulties subscale refers to what extent individuals encounter difficulties to reach specific outcomes online related to income, housing and healthcare. Second, the General Digital Difficulties construct refers to an individual's perception of frequently experiencing problems and questions when using a smartphone or computer in general. In others words, General Digital Difficulties could be considered as a state of general inconvenience in using a smartphone or computer. Finally, the subscale Worries about Future Digital Difficulties refers to individuals' insecurity or worries about not being able to use smartphones, computers, and other new technologies in the (near) future. Worrying is a common cognitive activity about everyday troubles, which can be based on specific threats and consequences, as well as on abstract threats and consequences [35]. As smartphones and computers are highly subjected to frequent updates and changes, a significant amount of people may associate this with high insecurity, in order to control this insecurity, it is plausible that individuals start to worry about the use of smartphones and computers in the future [55].

The Digital Difficulties Scale is of particular value for practitioners and policy makers, as this is the first instrument that investigates difficulties in reaching specific outcomes related to income, housing, and healthcare online. Today, research on inequality and poverty is undergoing a multidimensional turn, i.e. moving beyond an exclusive focus on income-centric forms of poverty to incorporate information from a wider set of dimensions that reflect the many different ways in which human life can be impoverished [56]. According to our view, experiencing digital difficulties can be viewed as an additional dimension of inequality, and as such our instrument can be incorporated in studies aimed to discern people who are at disadvantage. As answers to the items under Specific Digital Difficulties can be recalculated in a binary way (having no difficulty versus having difficulty), practitioners and policy makers are allowed to better estimate who may be disadvantaged by the abolishment of offline counters and services in the future (e.g., for civil affairs, banking or health consultations).

Notwithstanding its results, this study has some limitations that should be considered. First, although sample 3 was heterogeneous with regard to age, gender, educational level and employment status in Flanders (i.e. the Dutch speaking part of Belgium), the use of convenience samples limits the generalizability of our findings. Furthermore, due to our sampling procedure we may have specifically missed out those who are already disadvantaged and more hidden in society due to a lower income level, health status, social status, or migration background. Corroboration of our findings produced by representative data as well as data derived from disadvantaged groups would lend credibility to the findings. A second limitation of this study is that respondents where not questioned whether or not they perceive themselves as disadvantaged by the digital difficulties they encounter. In line with income poverty and material deprivation research, it is possible that individuals are considered as disadvantaged by the objective cutoff, although they do not perceive themselves as disadvantaged by their situation. Therefore future studies might investigate the predictive validity of having digital difficulties with regard to well-being and other life outcomes.

## Conclusion

In conclusion, it is important both for researchers, policy makers, and employers to understand who encounters difficulties in using a smartphone or computer, and therefore who is disadvantaged to fulfill basic needs in increasing digitizing societies. This concern has emerged as the use of smartphones and computers is being normalized (e.g., the (obligated) use of apps, websites or computer programs to make financial transfers, to claim benefits, or to search for job vacancies), despite that previous research demonstrated that a significant number of individuals lack smartphone, computer, or internet access, skills, or use variety [10,11,14,16].

Therefore, we developed the Digital Difficulties Scale measuring who is at a disadvantage to fulfill basic needs by having difficulties in using a smartphone or computer. We discerned three subscales: Specific Digital Difficulties or difficulties that individuals encounter if they have to reach specific outcomes online; General digital difficulties or encountering general inconvenience in using a smartphone or computer, and Worries about Future Digital Difficulties or feeling insecure about the use of digital technologies in the (near) future. Our instrument proved to have satisfying psychometric properties in terms of validity and reliability. However, further testing of the psychometric properties of the scale is recommended by conducting studies in different populations.

## Supporting information

**S1 Appendix. Full items of the Digital Difficulties Scale.**
(DOCX)

## Author Contributions

**Conceptualization:** Sarah Anrijs, Koen Ponnet.

**Formal analysis:** Sarah Anrijs, Koen Ponnet.

**Funding acquisition:** Sarah Anrijs, Koen Ponnet, Lieven De Marez.

**Investigation:** Sarah Anrijs.

**Methodology:** Koen Ponnet.

**Project administration:** Sarah Anrijs.

**Supervision:** Koen Ponnet.

**Writing – original draft:** Sarah Anrijs, Koen Ponnet.

**Writing – review & editing:** Sarah Anrijs, Koen Ponnet, Lieven De Marez.

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
