## [Decision Letter · Decision Letter 0]

17 Apr 2020

PONE-D-20-06232

Development and psychometric properties of the Digital Difficulties Scale (DDS): An instrument to measure who is disadvantaged to fulfill basic needs by experiencing difficulties in using a smartphone or computer

PLOS ONE

Dear Mrs. Anrijs,

Thank you for submitting your manuscript to PLOS ONE. After careful consideration, we feel that it has merit but does not fully meet PLOS ONE’s publication criteria as it currently stands. Therefore, we invite you to submit a revised version of the manuscript that addresses the points raised during the review process.

We would appreciate receiving your revised manuscript by Jun 01 2020 11:59PM. To enhance the reproducibility of your results, we recommend that if applicable you deposit your laboratory protocols in protocols.io, where a protocol can be assigned its own identifier (DOI) such that it can be cited independently in the future. For instructions see: http://journals.plos.org/plosone/s/submission-guidelines#loc-laboratory-protocols

We look forward to receiving your revised manuscript.

Kind regards,

Francesca Chiesi

Academic Editor

PLOS ONE

2. Please respond by return e-mail with an updated version of your manuscript to include your abstract after the title page.

Reviewers' comments:

Reviewer's Responses to Questions

**Comments to the Author**

1. Is the manuscript technically sound, and do the data support the conclusions?

Reviewer #1: Yes

Reviewer #2: Yes

2. Has the statistical analysis been performed appropriately and rigorously? 

Reviewer #1: Yes

Reviewer #2: Yes

3. Have the authors made all data underlying the findings in their manuscript fully available?

Reviewer #1: Yes

Reviewer #2: Yes

4. Is the manuscript presented in an intelligible fashion and written in standard English?

Reviewer #1: Yes

Reviewer #2: Yes

5. Review Comments to the Author

Reviewer #1: I have some quibbles with the population use. One could say that the initial population for item development is too limited (it seems to be mostly a Flemish library) and I am not that keen on using pre-existing panels (people are doing this for pay and might have strategies for their personal benefit). I also did not find the divergent/convergent validity that compelling. But the statistical justification is excellent. And I think a number of other researchers will be using this scale very soon and we will soon get a good idea of its generalizability. I believe the items were well conceived in a serious fashion. Most important I think this is a very timely survey as we try and understand inequalities and how they affect sustainability in our current historical moment. I think this paper should be published as quickly as possible (I am guessing the authors realized this and is one of the reasons they sent it to PlusOne). Maybe a little more justification for the populations used but I hope it is published as quickly as possible.

Reviewer #2: I would like to congratulate the auhtor for such interesting topic of research on a new questionnaire that measures Specific Digital Difficulties, General Digital Difficulties, and Worries about Future Digital Difficulties. The study seems relevant, well-performed and well-written. The results are interpreted carefully. However, there are some issues that I would like to address. I hope these could be of interest for them, and to improve the quality of the current manuscript.

Main points;1. As far as I am concerned, authors examine technology adoption. Even if there is innovation in the current work, this is a subject deeply studied in both applied and theoretical levels. Therefore,  I would avoid sentences such as "Until now, no adequate measure exists to assess ". I recommend extreme caution in the use of these phrases, as they can be misinterpreted. In this way, I ask the authors to delete or reformulate them.

2. The theoretical framework can be improved. I presonally missed some references on Attitudes Towards Technology Adoption. This is a suggestion and of course, I do not mean authors need to name each theoretical models. However, some reference to thiswould enrich the present manuscript. Likewise, I suggest including literature such as the following:

Boot, W. R., Roque, N., Charness, N. H., Rogers, W. A., Mitzner, T. L., Czaja, S. J., & Sharit, J. (2017). OLDER ADULT TECHNOLOGY PROFICIENCY AND TECHNOLOGY ADOPTION. Innovation in Aging, 1(Suppl 1), 1026.

Mitzner, T. L., Savla, J., Boot, W. R., Sharit, J., Charness, N., Czaja, S. J., & Rogers, W. A. (2019). Technology adoption by older adults: findings from the PRISM trial. The Gerontologist, 59(1), 34-44.

Moret-Tatay, C., & Murphy, M. (2019). Aging in the digital era. Frontiers in psychology, 10, 1815.

3. Which methods have authors employed for both EFA and CFA? Is there any rotation?

4. Please, discuss the nature of your polychoric scores and your data analysis strategy according to that

5. I missed a conclusion section

Minor points1. Please, include the abstract in the manuscript

2. What about the recommendation to the field? Which is the take-home message?

6. PLOS authors have the option to publish the peer review history of their article (what does this mean?). If published, this will include your full peer review and any attached files.

Reviewer #1: Yes: Michael Glassman

Reviewer #2: No

---

## [Author Response · Author response to Decision Letter 0]

7 May 2020

Reviewer #1: I have some quibbles with the population use. One could say that the initial population for item development is too limited (it seems to be mostly a Flemish library) and I am not that keen on using pre-existing panels (people are doing this for pay and might have strategies for their personal benefit). I also did not find the divergent/convergent validity that compelling. But the statistical justification is excellent. And I think a number of other researchers will be using this scale very soon and we will soon get a good idea of its generalizability. I believe the items were well conceived in a serious fashion. Most important I think this is a very timely survey as we try and understand inequalities and how they affect sustainability in our current historical moment. I think this paper should be published as quickly as possible (I am guessing the authors realized this and is one of the reasons they sent it to PlosOne). Maybe a little more justification for the populations used but I hope it is published as quickly as possible.

**answer to the reviewer**

**We would like to thank the reviewer for his/her nice comments. We understand the quibbles of the reviewer with our convenience samples, and recognize that in general probability sampling, in which individuals are chosen at random, is the preferred approach for scientifically conducted surveys. However, at the moment of conducting our study, we were unable to pursue a probability sample because of time and budget limitations. Therefore, future studies on digital difficulty should aim for probability samples, which may lend corroboration to our study findings. In addition to probability sampling, purposively non-probabilistic sampling can be valuable to evaluate inequalities, such as digital difficulty, among disadvantaged groups. Disadvantaged groups are often more affected by inequalities, but underrepresented in data collection as these groups are “hidden” in society and/or difficult to access. It is however the aim of our research group to administer our instrument among disadvantaged groups in the near future (i.e. after the corona period). 

**In order to meet the concerns of the reviewer, we have provided more information on the data collections. We provide more information about the recruitment of sample 1 and 2 on pages 9 and 12.With regard to sample 3, we inserted a new paragraph (page 12-13):

“Therefore, the scale was administered in a new sample of Flemish individuals (i.e., the Dutch speaking part of Belgium) between 18 and 64 years old.. Respondents were recruited by a professional research organization that has access to a panel of 300.000 Belgian individuals. Students and non-Dutch speaking persons were not eligible to participate. In total, 8000 panel members received an email with a short study description, informed consent, and invitation to participate, including a link to the survey. A stratified sampling procedure was applied in order to assure that the sample was heterogeneous. Based on the federal statistics of Belgium (www.statbel.fgov.be), we a priori stratified the data with regard to gender, age, employment status and educational degree, so that the proportion of the strata reflects the proportion of the Flemish population. At the moment that 1000 respondents had been reached in accordance with the interlaced strata, the survey link was closed by the professional research organization.”

**In addition, we explicitly mentioned the use of convenience sampling as a limitation in the manuscript (p22-23):

“Notwithstanding its results, this study has some limitations that should be considered. First, although sample 3 was heterogeneous with regard to age, gender, educational level and employment status in Flanders (i.e. the Dutch speaking part of Belgium), the use of convenience samples limits the generalizability of our findings Furthermore, due to our sampling procedure we may have specifically missed out those who are already disadvantaged and more hidden in society due to a lower income level, health status, social status, or migration background. Corroboration of our findings produced by representative data as well as data derived from disadvantaged groups would lend credibility to the findings.”

Reviewer #2: I would like to congratulate the author for such interesting topic of research on a new questionnaire that measures Specific Digital Difficulties, General Digital Difficulties, and Worries about Future Digital Difficulties. The study seems relevant, well-performed and well-written. The results are interpreted carefully. However, there are some issues that I would like to address. I hope these could be of interest for them, and to improve the quality of the current manuscript.

**answer to the reviewer**

**We would like to thank the reviewer for his/her nice comments.

Main points;

1. As far as I am concerned, authors examine technology adoption. Even if there is innovation in the current work, this is a subject deeply studied in both applied and theoretical levels. Therefore, I would avoid sentences such as "Until now, no adequate measure exists to assess ". I recommend extreme caution in the use of these phrases, as they can be misinterpreted. In this way, I ask the authors to delete or reformulate them.

**answer to the reviewer**

**Thank you for this comment. In the revised version of the paper with track changes, we have reformulated two phrases and we have deleted another one (lines 23, 68-70, 419-421).

2. The theoretical framework can be improved. I personally missed some references on Attitudes Towards Technology Adoption. This is a suggestion and of course, I do not mean authors need to name each theoretical models. However, some reference to this would enrich the present manuscript. Likewise, I suggest including literature such as the following:

Boot, W. R., Roque, N., Charness, N. H., Rogers, W. A., Mitzner, T. L., Czaja, S. J., & Sharit, J. (2017). OLDER ADULT TECHNOLOGY PROFICIENCY AND TECHNOLOGY ADOPTION. Innovation in Aging, 1(Suppl 1), 1026.

Mitzner, T. L., Savla, J., Boot, W. R., Sharit, J., Charness, N., Czaja, S. J., & Rogers, W. A. (2019). Technology adoption by older adults: findings from the PRISM trial. The Gerontologist, 59(1), 34-44.

Moret-Tatay, C., & Murphy, M. (2019). Aging in the digital era. Frontiers in psychology, 10, 1815.

**answer to the reviewer**

**We agree with the reviewer that attitudes and adoption are often studied within digital inequality research. Therefore we adapted our synthesis of digital inequality research in the introduction (lines 60-67), and added the suggested references, as these where very helpful for our synthesis. 

3. Which methods have authors employed for both EFA and CFA? Is there any rotation?

**answer to the reviewer**

**Thank you for this valuable question, we added this information in the result section of our manuscript as follows:

Page 16: “First, we conducted an EFA using principal axis factoring with varimax rotation.”

Page 17: “As our data were non-normally distributed, results were obtained with the maximum likelihood mean adjusted.”

4. Please, discuss the nature of your polychoric scores and your data analysis strategy according to that

**answer to the reviewer**

**We treated the constructs as continuous variables, and discuss them accordingly.

5. I missed a conclusion section

**answer to the reviewer**

**In the revised manuscript, we have added a separate section heading “Conclusion”

Minor points

1. Please, include the abstract in the manuscript

**answer to the reviewer**

**This comment has been solved.

2. What about the recommendation to the field? Which is the take-home message?

**answer to the reviewer**

**Thank you for this useful comment, a clear recommendation for the field was currently lacking. We added an implication/recommendation in the revised version (page 22): 

“The Digital Difficulties Scale is of particular value for practitioners and policy makers, as this is the first instrument that investigates difficulties in reaching specific outcomes related to income, housing, and healthcare online. Today, research on inequality and poverty is undergoing a multidimensional turn, i.e. moving beyond an exclusive focus on income-centric forms of poverty to incorporate information from a wider set of dimensions that reflect the many different ways in which human life can be impoverished [56]. According to our view, experiencing digital difficulties can be viewed as an additional dimension of inequality, and as such our instrument can be incorporated in studies aimed to discern people who are at disadvantage. As answers to the items under Specific Digital Difficulties can be recalculated in a binary way (having no difficulty versus having difficulty), practitioners and policy makers are allowed to better estimate who may be disadvantaged by the abolishment of offline counters and services in the future (e.g., for civil affairs, banking or health consultations).“

---

## [Editor Report · Decision Letter 1]

15 May 2020

Development and psychometric properties of the Digital Difficulties Scale (DDS): An instrument to measure who is disadvantaged to fulfill basic needs by experiencing difficulties in using a smartphone or computer

PONE-D-20-06232R1

Dear Dr. Anrijs,

We are pleased to inform you that your manuscript has been judged scientifically suitable for publication and will be formally accepted for publication once it complies with all outstanding technical requirements.

With kind regards,

Francesca Chiesi

Academic Editor

PLOS ONE

---

## [Editor Report · Acceptance letter]

21 May 2020

PONE-D-20-06232R1 

Development and psychometric properties of the Digital Difficulties Scale (DDS): An instrument to measure who is disadvantaged to fulfill basic needs by experiencing difficulties in using a smartphone or computer 

Dear Dr. Anrijs:

I am pleased to inform you that your manuscript has been deemed suitable for publication in PLOS ONE. Congratulations! Your manuscript is now with our production department. 

With kind regards,

on behalf of

Dr. Francesca Chiesi 

Academic Editor

PLOS ONE